# Molecular mechanism of proton-coupled ligand translocation by the bacterial efflux pump EmrE

**Jakub Jurasz**[1], **Maciej Bagiński**[1,2], **Jacek Czub**[2,3], **Miłosz Wieczór**[3,4] *

**1** Department of Pharmaceutical Technology and Biochemistry, Gdansk University of Technology, Gdansk, Poland, **2** BioTechMed Center, Gdansk University of Technology, Gdansk, Poland, **3** Department of Physical Chemistry, Gdansk University of Technology, Gdansk, Poland, **4** Molecular Modeling and Bioinformatics Group, IRB Barcelona, Barcelona, Spain

* milwiecz@pg.edu.pl

**Data Availability Statement:** All relevant data are within the manuscript and its Supporting information files.

**Funding:** The author(s) received no specific funding for this work.

## Abstract

The current surge in bacterial multi-drug resistance (MDR) is one of the largest challenges to public health, threatening to render ineffective many therapies we rely on for treatment of serious infections. Understanding different factors that contribute to MDR is hence crucial from the global "one health" perspective. In this contribution, we focus on the prototypical broad-selectivity proton-coupled antiporter EmrE, one of the smallest known ligand transporters that confers resistance to aromatic cations in a number of clinically relevant species. As an asymmetric homodimer undergoing an "alternating access" protomer-swap conformational change, it serves as a model for the mechanistic understanding of more complex drug transporters. Here, we present a free energy and solvent accessibility analysis that indicates the presence of two complementary ligand translocation pathways that remain operative in a broad range of conditions. Our simulations show a previously undescribed desolvated apo state and anticorrelated accessibility in the ligand-bound state, explaining on a structural level why EmrE does not disrupt the pH gradient through futile proton transfer. By comparing the behavior of a number of model charged and/or aromatic ligands, we also explain the origin of selectivity of EmrE towards a broad class of aromatic cations. Finally, we explore unbiased pathways of ligand entry and exit to identify correlated structural changes implicated in ligand binding and release, as well as characterize key intermediates of occupancy changes.

## Author summary

EmrE is a prototypical bacterial multidrug transporter (MDR) that confers resistance to drugs and antiseptics. Due to its structural simplicity, its mechanism of ligand recognition and translocation are relevant for a wide class of transporters. This proton-coupled antiport expels aromatic cations from the cytoplasm using the alternating access mechanism, achieving impressive levels of efficiency and robustness. Our protonation-specific free energy profiles, Grotthuss wire analyses and equilibrium simulations show how a

**Competing interests:** The authors have declared
that no competing interests exist.

deceivingly simple system can exchange ions with robustness and precision, hopefully
inspiring rational efforts to design new MDR inhibitors.

## 1 Introduction

Increasing bacterial resistance to many first-line drugs and antiseptics has long been seen as a
global threat to public health. One of the main culprits, and hence a possible therapeutic target,
are the bacterial multidrug transporters (MDR) [1–4]. These transporters play an important
role in antibiotic resistance of several important human pathogens, such as *Mycobacterium
tuberculosis*, *Streptococcus pneumoniae*, *Pseudomonas aeruginosa* or *Vibrio cholerae* [5–7].
Therefore, detailed knowledge about the molecular basis of drug recognition and transport by
multi-drug transport systems is needed to develop inhibitors blocking the MDR transporters
or new, non-extruded antibiotics. To work effectively, these proteins must recognize the uni-
versal characteristics of potentially toxic compounds and then expel them from the cytoplasm
in an energy-dependent manner [8–11]. Many transporters specifically target polyaromatic
cations as they make up a significant fraction of both natural and man-made antibiotics, anti-
septics or anti-pathogens (e.g., benzalkonium chloride, ampicillin, erythromycin or tetracy-
cline) [12–15].

A representative broad-specificity transporter of this kind is EmrE, a multidrug transporter
from *Escherichia coli* [16–20]. It functions as an antiparallel homodimer, and is a prototypical
member of the small multidrug resistance superfamily (SMR) [21–23]. As one of the smallest
functional transporters, consisting only of 110 amino acids, it has become a model example for
basic and applied research [24]. Many results can be directly applied to orthologous systems
such as emrEPae, found in *P. aeruginosa* [25], a bacterium which according to the World
Health Organization is responsible for serious hospital infections in immunocompromised
patients and is highly resistant to antibiotic treatment. To date, the transport mechanism of
EmrE has not been unambiguously established on the atomic level, nor has a high-quality X-
ray or cryo-EM structure been obtained experimentally, a common issue among small flexible
proteins. However, the available low-resolution structure [26] has recently been refined using
molecular modeling, enabling further structure-based investigations that revealed the inner
workings of the enzyme in finer mechanistic detail [27, 28], while a high-quality NMR struc-
ture was published this year [29], providing a more reliable view of specific side chain confor-
mations. Under physiological conditions, EmrE uses the proton motive force (PMF) to act as
an antiporter; under specific conditions, though, it was shown to behave as a symporter,
highlighting its functional versatility [30]. Located in the bacterial inner membrane, its task is
to extrude positively charged, aromatic molecules from the cytoplasm in exchange for up to
two periplasmic protons, thus ensuring bacterial resistance to various toxic compounds [30].
This transporter activity crucially depends on two central E14 residues that form the ligand
binding site [31–34]. NMR studies have shown that in the presence of ligand, the two asym-
metrical subunits of the homodimer swap conformations, which reportedly coincides with a
change in the protonation state and ligand release [35] (see Fig 1). Hence during the transport
cycle, the binding site initially facing the high-pH cytoplasm becomes exposed to the low-pH
periplasm, allowing a proton to reach the central E14 residues [33, 36]. It has been proposed
that the resulting side chain protonation reduces affinity for the charged ligand, thereby allow-
ing for the completion of the cycle through ligand dissociation and return to the apo state [30].

In this work, we use molecular dynamics (MD) simulations to propose a high-resolution
mechanistic description of proton-coupled ligand transport of EmrE based on protonation-

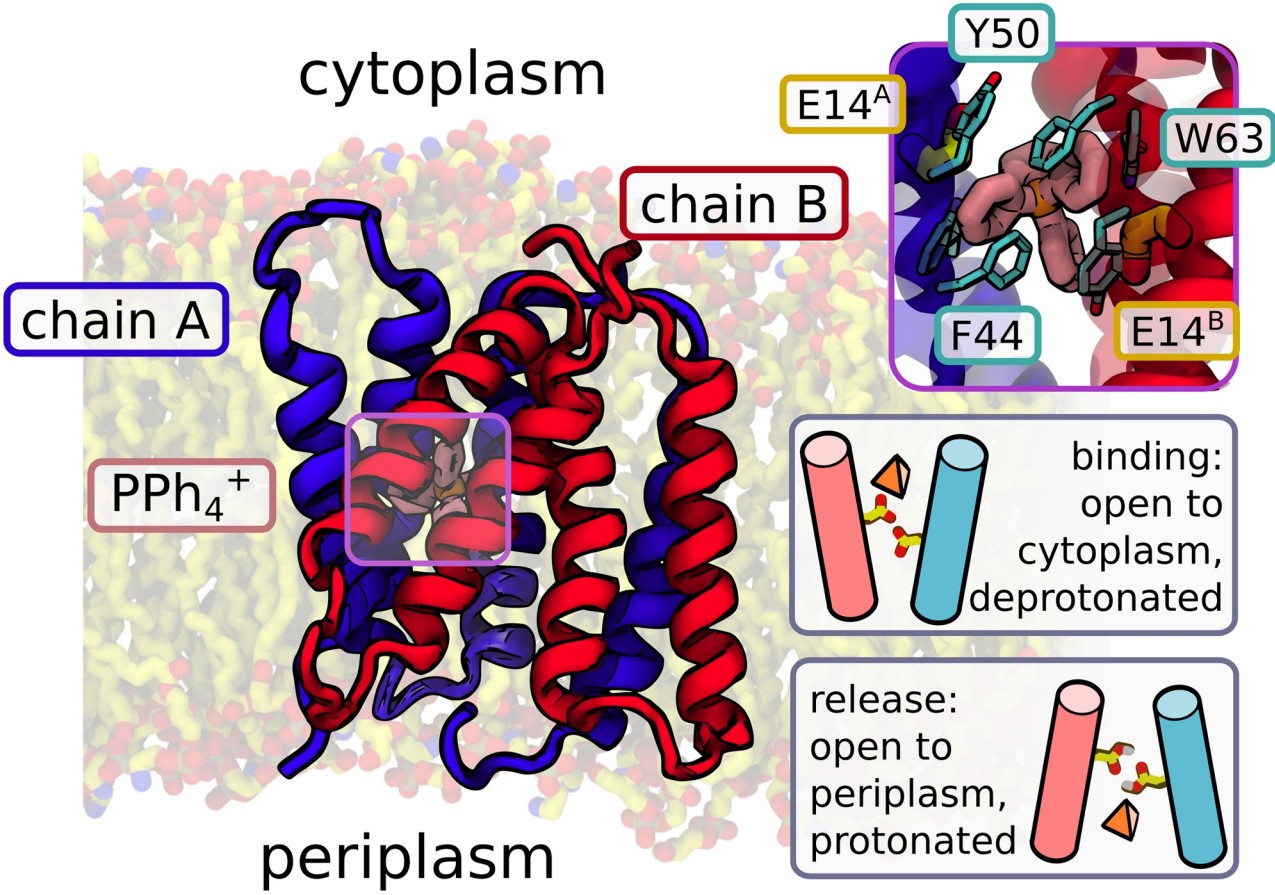

**Fig 1. Structure of EmrE.** The protein is an asymmetric antiparallel homodimer with a single ligand binding site, constituted by two glutamates (E14$^A$ and E14$^B$, contributed by both chains) and lined with aromatic residues (see inset). The dominant "alternating access" model of antiport assumes that deprotonated EmrE captures a hydrophobic ligand from the cytoplasm, switches to a conformation open towards the periplasm, and releases the ligand when protons from the low-pH conformation protonate the central glutamates. Protonated EmrE then switches back to the initial conformation, releasing two protons to the cytoplasm.

dependent free energy profiles and water accessibility analyses, building on recent computational studies and available cryo-EM data. We also investigate the mechanisms of substrate selectivity, showing how the properties of various ligands dictate strategies for efficient capture of aromatic cations. Finally, we explore the occupancy changes relevant to the translocation cycle—ligand binding, ligand release and the consecutive desolvation of the apo state—using weighted ensemble and equilibrium simulations to identify structural features crucial to the robust functioning of this prototypical transporter.

## 2 Results

### 2.1 Cryo-EM densities help obtain a more reliable initial structure

We used the EmrE protein structure refined by Ovchinnikov *et al.* [28] as a starting point for further refinement. After placing the structure in a model bacterial membrane, the system was first equilibrated for 1 $\mu$s (see S1 Fig). Then, using correlation-based cryo-EM fitting, the protein was restrained to match its electron density map published previously [26]. In this process, a 'forward' model of the electron density is created by Gaussian-smoothing the atomistic

structure, and forces are applied so as to minimize the cross-correlation between the simulated and actual density. Thanks to this, the spatial arrangement of the protein helices closely matches that of the PDB entry 3B5D (see S2 Fig), increasing the reliability of our initial structure. We then created a relaxed "apo" form by removing the ligand from this system and simulating it under equilibrium conditions for 1.75 $\mu$s. Recently, a new NMR-based structure was deposited (PDB 7JK8), and we report its similarity to existing models (PDB 3B5D, our refined structure and the structure obtained by Ovchinnikov V [37]) in S3 Fig.

## 2.2 Mechanism of ligand translocation is dictated by thermodynamics of PPh$_4^+$ binding

In past studies, the ligand translocation pathway of EmrE was broadly characterized in biochemical terms and at timescales accessible to NMR experiments, but these insights into the dynamics of binding were never directly explained on a structural level [30]. Hence, to gain a higher-resolution mechanistic understanding of the transport cycle of EmrE, we employed fully atomistic free energy simulations aimed at reconstruction of thermodynamics of PPh$_4^+$ binding at different protonation states of the central glutamates: E14$^A$ and E14$^B$. The resulting consensus free energy profiles, showed in Fig 2B, are supported by 30 $\mu$s of sampling each, and obtained as an average of results from three independent umbrella sampling and metadynamics runs (see Methods and S4 Fig for details). From the depths and positions of free energy minima, the profiles allow for reconstruction of a sequence of events—including protonation changes and the "domain-swap" conformational transition—that lead to ligand translocation driven by proton gradient.

Upon binding to a deprotonated EmrE dimer (solid orange line in Fig 2B–2D), PPh$_4^+$ encounters a series of metastable states, from the entry point at the bilayer surface level (z = 2.2 nm) through two local minima at z = 1.5 and 0.8 nm, to end up stably bound in the free energy basin next to the protein center, between z = -0.1 and 0.5 nm. From here, three different events can follow, with different mechanistic implications: (1) a conformational transition that swaps the domain labels and exposes the binding site to the periplasm; (2) protonation of E14$^A$; or (3) protonation of E14$^B$. To narrow down the scope of possibilities, we calculated pK$_a$ values of the two glutamates in a ligand-bound state using equilibrium alchemical free energy simulations. This approach yielded a pK$_a$ for E14$^B$ higher by more than 2 units than that of E14$^A$ (see S1 Table), in agreement with the assignment made in NMR experiments [30]. Even though absolute values of pK$_a$ in proteins cannot be calculated with precision higher than 1.5–2 units [38], and our estimate of absolute pK$_a$s is indeed off by at least 1 unit, this qualitative consistency shows that our model correctly captures the preferential protonation of E14$^B$.

To distinguish between the remaining two options—conformational change or protonation of E14$^B$—we calculated the fraction of Grotthuss water wires [39] connecting the glutamate side chains to bulk solvent from each side, cytoplasmic (high-pH) and periplasmic (low-pH), reasoning that E14$^B$ will only be protonated first if it can access protons from the periplasmic side with a PPh$_4^+$ ligand bound. As shown in Fig 2A, this is not the case, since E14$^B$ can freely exchange protons with the cytoplasmic side (dashed red line) but not with the periplasm (solid red line); hence the conformational change should occur as the next step, as in the pathway depicted in Fig 2C.

Surprisingly, however, E14$^A$ turned out to be capable of exchanging protons with the periplasm in the presence of a bound ligand (grey solid line), suggesting that under specific conditions (pH of the periplasm lower than the pK$_a$ of E14$^A$) this residue can still be protonated before the completion of the conformational change, as shown schematically in S5 Fig. As we

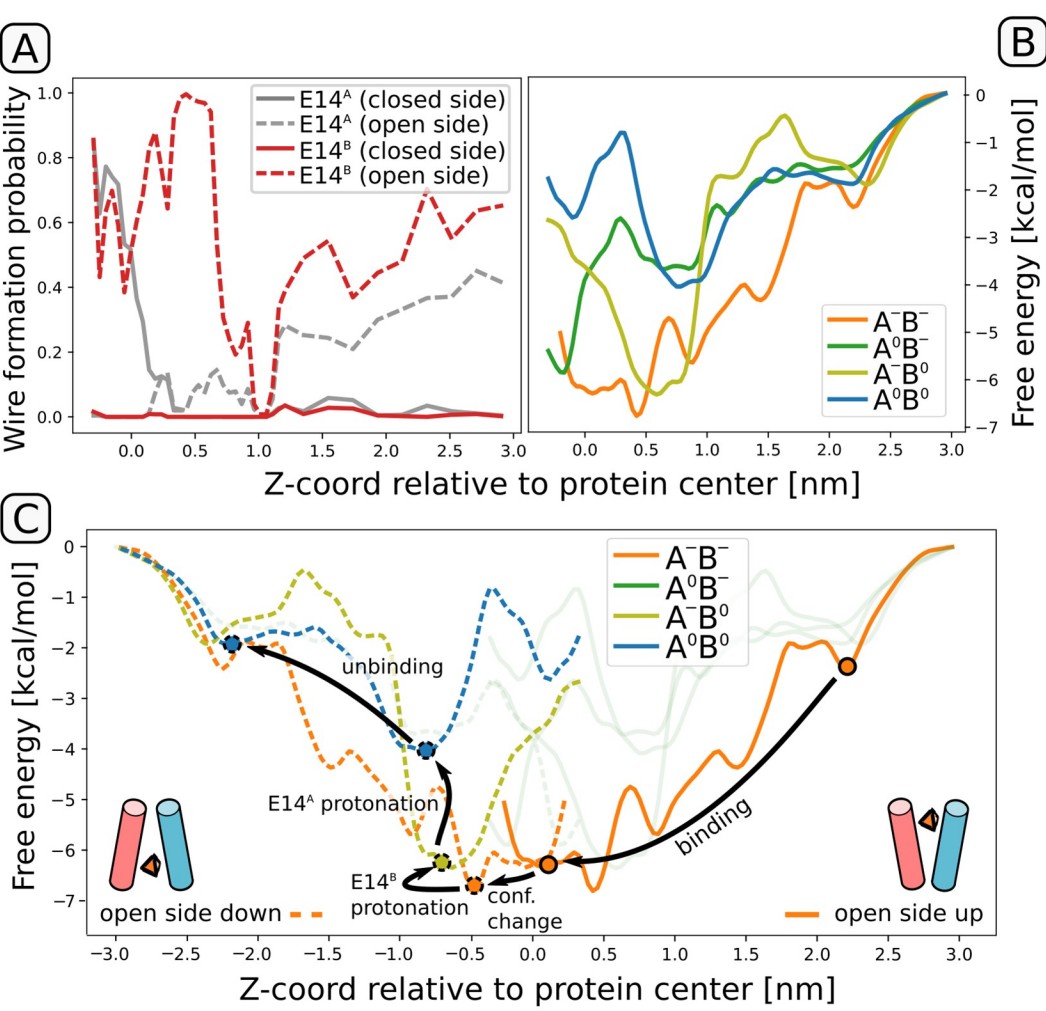

**Fig 2.** (A) Probabilities of Grotthuss wire formation. For both glutamates, the fraction of frames in which a continuous wire was found between the carboxyl group and bulk solvent is plotted as a function between the ligand position and the probability calculated as an average from each umbrella sampling window. Note that "open" and "closed" side might refer to either cytoplasm or periplasm, depending on the current conformational state of EmrE. (B) Free energy profiles for ligand entry in the four possible protonation states. The order of affinities is consistent with the intuitive notion that each protonation event decreases affinity for the positively charged ligand. (C) The most thermodynamically plausible mechanism of proton-coupled antiport, postulated based on wire probabilities from panel A and protonation-dependent free energy profiles from panel B (note the dashed profiles are just reflected about Z = 0 and correspond to a state after the conformational change). The mechanism assumes $E14^B$ protonates more eagerly in the ligand-bound state, as shown by experiments and simulations. Note that upon a symmetric transition, the $pK_a$ of $E14^B$ becomes the $pK_a$ of $E14^A$ and vice versa. For a dynamic version of the figure, see S1 Movie. For the discussion of an alternative pathway, cf. S5 Fig.

shall show, both sequences of events lead to slightly distinct but mechanistically viable translocation pathways.

Assuming that the conformational change occurs right after ligand entry, the dimer is now open to the periplasm, and $E14^B$ can easily accept a proton, an event corresponding to a jump from the orange to the lime free energy profile in Fig 2C. Accordingly, the equilibrium position of the ligand shifts towards exit (z = -0.6 nm), where acceptance of another proton is feasible, as shown by the dashed grey line of Fig 2A. Upon second protonation, the system transitions to the blue profile corresponding to a doubly protonated state, where only two minor free

energy barriers, of ca. 2 kcal/mol each, separate $PPh_4^+$ from full dissociation of from the protein. This complete translocation pathway is illustrated in S1 Movie.

If in turn the proton is instead first accepted by E14$^A$ at sufficiently low periplasmic pH, the ligand will preferentially progress towards more negative values of z (minimum at -0.2 nm, green profile in Fig 2B). Once the conformational change occurs, the symmetrical change of labels "converts" E14$^A$ into E14$^B$, resulting in a situation identical to the one described above. We note that this dual pathway might represent an evolutionary optimization, allowing the protein to operate at different pH differences, as long as the lower periplasmic pH is capable of protonating both glutamates.

Notably, although the profiles capture the intuitive order of affinities in which each consecutive protonation event decreases the electrostatic attraction between the negatively charged binding site and the aromatic cation, the exact affinities systematically underestimate the experimentally determined values. For the deprotonated and doubly protonated variants, our profiles reach minima of -7 and -4 kcal/mol instead of -11 and -6 kcal/mol [30, 33], likely reflecting the fact that our free energy simulations do not reach the full apo state. However, we point out that it is the structural adaptations of the binding site and its neighborhood that confers efficiency and robustness of the mechanism. In particular, while a watertight dimer interface prohibits Grotthuss wire formation between the closed side and E14$^B$ at all times (Fig 2A), E14$^A$ alternates between connection to the closed (z<0.5 nm) and open (z<0.5 nm) side, thereby enabling adaptability while preventing futile proton transfer. This might address a similar concern regarding the presence of spurious "leaky" states in simulations performed by Vermaas *et al.* [27].

## 2.3 Binding site desolvation in the apo form prevents proton leaks

We show above that in the ligand-bound state, E14$^A$ and E14$^B$ are connected to two different proton reservoirs, explaining how EmrE prevents proton leaks in the ligand-bound state. However, a previous simulational study [28] found that the apo form maintains the Grotthuss-like connection between the binding site glutamates and the bulk solvent on a sub-microsecond scale, one that would raise the possibility of "futile" proton transport as the apo form undergoes spontaneous conformational changes.

To verify this finding, we performed a longer simulation of the apo form, extending them to almost 2 $\mu$s, and analyzed the resulting trajectory for the presence of continuous hydrogen-bonded water wires. This equilibrium simulation revealed that following the (unphysiologically abrupt) removal of the ligand, EmrE initially remains in a conformation close to the ligand-bound one. The initial reconfiguration of contacts is rather modest, and results in alternating changes in the accessibility of the binding site (Fig 3C). After 1.2 $\mu$s, however, a concerted rotameric change of W63$^B$ and M21$^B$ created a hydrophobic barrier around the binding site, so that all Grotthuss wires connected to E14$^A$ and E14$^B$ abruptly disappeared. After additional 0.5 $\mu$s, the binding site remained inaccessible to bulk water, with 8 water molecules stably trapped inside the central cavity (Fig 3D). This transition should be easily reversible when the hydrophobic ligand approaches the protein, indicating that the key structural changes between the apo and ligand-bound forms are fast enough to avoid unwanted proton leaks. We need to note, though, that the sole fact of measuring pK$_a$ values for the apo form [40] implies that such spontaneous reopening has to be feasible (so that the protons can reach the central glutamates), albeit likely on a much slower timescale than the one relevant for transfer. Indeed, a comparison of ensemble distributions of interresidue distance measured using spin-label based DEER showed that the apo form has a much less defined structure, hinting at a more complex conformational landscape of the ligand-free form [41].

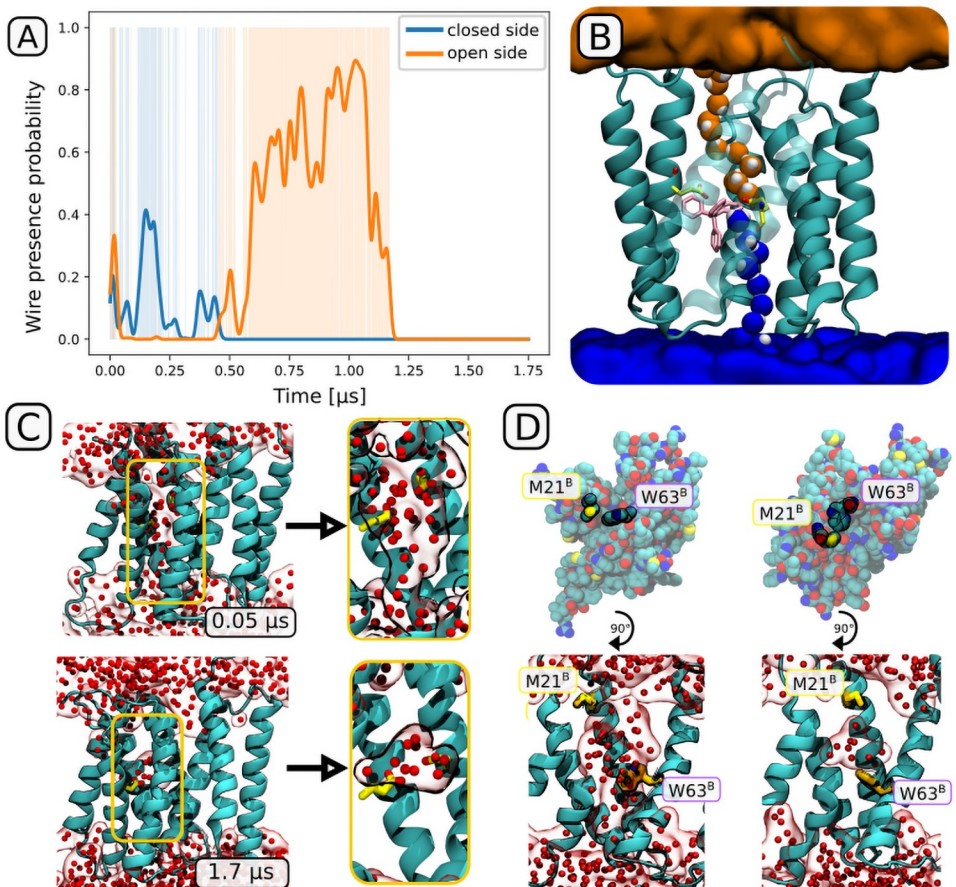

**Fig 3.** (A) Presence of the Grotthuss wires during the 1.75 μs equilibrium simulations of the apo state. At 1.2 μs, the wires disappear permanently. Plotted as a function between the time of simulation and the probability calculated as a Gaussian-smoothed wire presence function. (B) A visualization of wires extending from the binding site to the open (orange) and closed (blue) side, with the $PPh_4^+$ ligand shown in pink and the two glutamates in yellow. (C,D) Loss of solvation of the ligand binding channel, shown as a comparison between the initial and final structures. (C) Disappearance of the water wires, with only several (8 in this case) water molecules remaining trapped in the binding site cavity. (D) Dominant role of the M21$^B$ "plug" and the W63$^B$ "valve" in the desolvation of the binding channel (see also S2 Movie).

## 2.4 Solvent-exposed surface of EmrE confers selective affinity for charged aromatic species

EmrE exhibits broad specificity for aromatic cations, compounds whose antibacterial properties often stem from their non-specific interaction with DNA and/or RNA. To investigate the structural basis for this selectivity, as well as provide a clear mechanism of substrate recognition, we studied the behavior of membrane-embedded EmrE in the apo form in the presence of four different ligands, three consisting of four phenyl groups bound to a central atom (cationic $PPh_4^+$, neutral $SiPh_4$ and anionic $BPh_4^-$) and one non-aromatic yet hydrophobic cation ($NMe_4^+$). Using pure (protein-free) bilayer systems, we first verified that all charged ligands preferentially reside in the aqueous phase near the bilayer surface, while the neutral tetraphenylsilane shows high affinity for the bilayer interior (see S6 Fig). This indicates that the cationic substrates have to be bound directly from the cytoplasm-bilayer interface—a variant of the popular "molecular vacuum cleaner" mechanism [42], where instead of being captured from

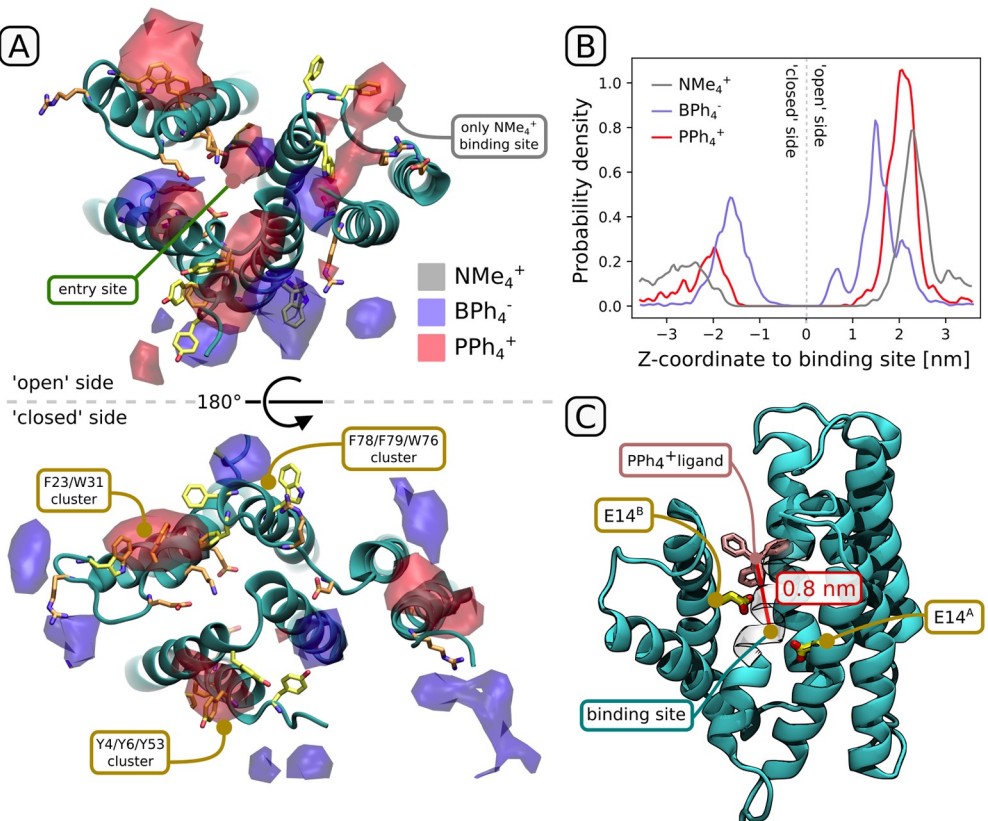

**Fig 4. Binding preference of charged and/or aromatic ligands to the solvent-exposed surface of EmrE.** (A) 3D densities of the central atom near the solvent-exposed surface of EmrE, looking from the "open" (top) and "closed" (bottom) side. Aromatic side chains exposed at the surface are shown in yellow, charged in orange. In each set of simulation, 10 ligands were initially placed in the water phase and allowed to interact with the protein. (B) Ligand density as in (A), but projected on the z-axis. Note that $BPh_4^-$ entered into spaces between the protein and the lipid bilayer, occasionally achieving z<0.5 nm but outside of the main ligand-binding channel. (C) A sample structure of ligand-bound EmrE in which the ligand spontaneously passed from bulk solvent to z = 0.8 nm (see extended simulations in S7 Fig).

inside a membrane leaflet, the charged hydrophobic ligand preferentially adsorbs at the exposed surface of the membrane protein.

Indeed, in the presence of membrane-embedded EmrE, a large fraction of charged aromatic ligands accumulated at the water-protein interface, with a slight preference for the "open" side of the protein (Fig 4A and 4B). In turn, the non-aromatic hydrophobic cation ($NMe_4^+$) showed virtually no preference for the protein surface, even though it was found to be an exceptionally good substrate for transport [43]. This indicates that regardless of the ligand's affinity for the central binding site, aromatic ligands are strongly selected for during initial binding, likely facilitating their expulsion even at low concentrations. Several low-affinity binding sites at the protein surface are directly responsible for this effect, mostly coinciding with surface clusters of aromatic or charged residues. In fact, the observed high ligand densities are almost entirely explainable by the presence of two aromatic residue clusters (Y4/Y6/Y53 and F23/W31/F78/F79/W76, the latter split in two at the "open" side of the dimer; yellow in Fig 4A) and/or charged residues (orange in Fig 4A) on the protein surface.

While it is surprising that both $PPh_4^+$ and $BPh_4^-$ accumulated at the protein surface to a similar extent, in all replicas only the aromatic cation managed to properly enter the channel

leading to the binding site, eventually reaching a distance of 0.8 nm from the center of the protein (S8 Fig). This hints at the presence of another selectivity filter, preventing aromatic anions from accessing the hydrophobic channel leading to the binding site even before they are repelled by the pair of negatively charged E14 residues.

However, no single charged or hydrophilic residue seems to single-handedly confer this preference, as the channel entrance is lined with hydrophobic side chains. To observe any differences between an anionic and cationic ligand in a more controlled setting, we extracted 3 channel-bound ligand structures (z-distance to binding site between 1.0 and 1.2 nm) from the equilibrium runs and performed separate 1 $\mu$s simulations of both $PPh_4^+$ and $BPh_4^-$, running it "as it is" or swapping the charges but keeping the geometry (depending on which system the structure came from; see S7 Fig). Out of the 3 $PPh_4^+$ systems (yellow/orange lines in S7 Fig), all remained stably bound to the channel entrance, mostly entering deeper (up to 0.8 nm) into the channel (see Fig 4C and free energy minima in Fig 2B); simultaneously, the least strongly bound $BPh_4^-$ dissociated in one of the systems, while the other two progressed deeper into the binding channel similarly to their cationic counterparts (blue/purple lines in S7 Fig). We hence conclude that the selectivity at the channel entrance is weak and might instead stem from the initial partial desolvation of the ligand entering the channel, as would be suggested by the different costs of desolvation, known from experimental data to differ by more than 2 kcal/mol between $PPh_4^+$ and $BPh_4^-$ [44]. Further down the channel, selectivity for cations would be trivially conferred due to the electrostatic effect of deprotonated (and hence negatively charged) $E14^A$ and $E14^B$; indeed, the effect of electrostatic attraction on the ligand affinity of the binding site is easily observable in the free energy analysis (cf. orange/blue profiles in Fig 2).

## 2.5 Ligand entrance and exit are associated with well-defined structural changes

Despite the effectively multi-$\mu$s timescale explored in our equilibrium spontaneous entry simulations, we did not observe a complete ligand entrance in which $PPh_4^+$ would sample a full pathway connecting the apo and ligand-bound states. To provide a realistic description of such complete transitions on the atomistic level, we turned to the Weighted Ensemble approach [45]—implemented in WESTPA—to generate unbiased trajectory ensembles of $PPh_4^+$ entrance into the unprotonated binding site, as well as $PPh_4^+$ release from the doubly protonated binding site. In doing so, we expected to observe key structural changes associated with ligand binding and release, and to identify possible intermediate steps along both pathways, potentially generalizable to other EmrE substrates.

Using the WESTPA workflow, we were able to generate successful ligand entry and escape trajectories with a length of up to 10 and 4 ns, respectively (see S3 Movie). Since these simulations represent physical but low-probability instances of such transitions, we ran equilibrium simulations from selected end state configurations to verify that the resulting configurations remain stable over time. In fact, in the shortest binding trajectories the protein clearly failed to adjust to the presence of the ligand, as the ligand quickly escaped the binding site in a plain equilibrium simulation, returning to an intermediate position at z = 0.8 nm. In subsequent trajectories, however, the ligand remained stably bound after entry for at least 100 ns. A visual inspection of differences between these two classes of trajectories indicate that reorganization of E14 and W63 is crucial for the stable accommodation of $PPh_4^+$ in the binding site, as shown in S8 Fig.

To identify structural changes accompanying changes in ligand occupancy in a systematic manner, we used the longest available entry/escape trajectories, merged with follow-up

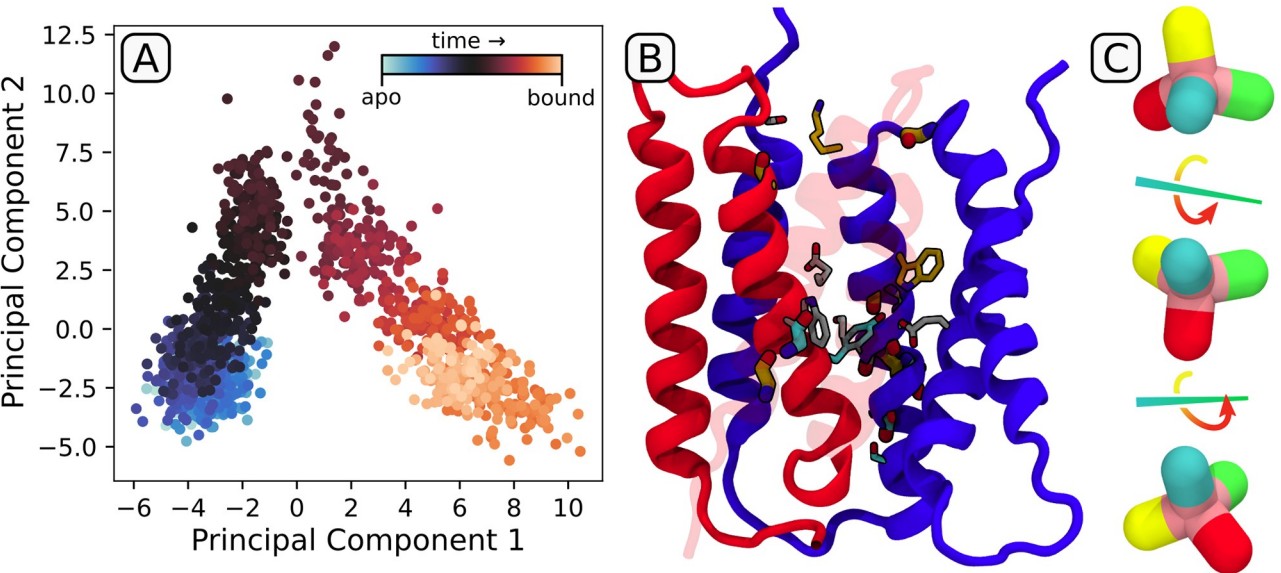

**Fig 5. Spontaneous ligand entry trajectory from weighted ensemble simulations, extended with a 100 ns equilibrium trajectory.** (A) Projection of the trajectory on the two main principal components describing correlated changes in H-bonding patterns. (B) Residues involved in the definition of PC1. Thick, H-bonds involve the backbone; thin, H-bonds engage the side chain. Residues were color-coded to show if they gain (orange), change (gray) or lose (cyan) H-bonds upon ligand binding. (C) Schematic description of the "tumbling" mechanism of ligand binding and unbinding, with individual colors corresponding to distinct phenyl groups of $PPh_4^+$.

equilibrium ones, to separately perform principal component analysis (PCA) on protein-protein hydrogen bonds. The resulting projection of the ligand entry trajectory on the two principal components, shown in Fig 5A, allows to distinguish between two kinds of correlated structural alterations: one associated with the transition from the apo to the bound form (PC1), and another connected to a transient and reversible reorganization of the binding channel (PC2). The residues contributing the most to PC1, shown in Fig 5B, cluster in several groups, mostly surrounding the binding site. In particular, the contacts of $Y60^A$ with $E14^A$ and $W63^B$ with $E14^B$, present in the apo state and disrupted by the incoming ligand, are partially compensated by new H-bonds between $E14^A$ and $W63^A$ and $S43^B$ with $E14^B$. Simultaneously, backbone H-bonding in a region of transmembrane helix 3 (TM3) stretching from $G65^A$ to $I71^A$ undergoes rearrangement due to formation of a kink in the exceptionally elastic 65GVG67 region. This kinking was indeed deemed important for ligand binding and the rate of conformational change, as concluded from past mutagenesis, NMR, EPR and cryo-EM studies [26, 29, 46, 47], and might provide an easy way of storing excess free energy for the subsequent ligand release. Finally, minor rearrangements occur at the entry site—$S75^B$ switching from an intermolecular H-bond with $S105^A$ to an intramolecular one with the backbone of a neighboring $I75^B$—reflecting a more open conformation of the binding channel in the bound than in the apo state.

Due to a much slower timescale of the bound→apo transition, the structural changes in the exit trajectories are much less pronounced and mostly map to spatially distinct regions, much less informative than the entry pathway described above (see S9 Fig for the corresponding PCA projection plot). However, both entry and exit trajectories highlight an interesting feature of the ligand progression through the binding channel, during which individual phenyl groups of $PPh_4^+$ "tumble" in a discrete way (see Fig 5C and S2 Movie). This step-wise movement explains the presence of several free energy minima in Fig 2B, and is enabled by correlating

these "tumbles" with rotations of aromatic side chains lining the binding channel (inset of Fig 1), in line with observations from a previous simulation study [48]. This suggests that apart from increasing the affinity for aromatic ligands and sealing the access of water in the apo form, the presence of aromatic residues in the binding site might also serve to accelerate the kinetics of occupancy changes.

## 3 Conclusions

In this contribution, we provide the first fully dynamic atomistic description of the proton-coupled antiport activity of EmrE. Our extensive free energy simulations revealed the existence of two translocation pathways, differing slightly in the order of events depending on the relationship between instantaneous $pK_a$ values of the central site glutamates and the periplasmic pH. In particular, by tracing the formation of ordered Grotthuss water wires, we observed that ligand entrance enables proton transfer to the low-$pK_a$ glutamate if the periplasm is sufficiently acidic, and otherwise the protonation of the other glutamate is enabled after the conformational change. Yet, both pathways push the ligand towards exit, so that after another protonation event the ligand is poised for release. Since both pathways are functionally indistinguishable, we speculate that this versatility might provide an adaptation to slightly variable environmental conditions. We also show how precise anticorrelation of accessibility prevents futile proton transport in the ligand-bound form. In the apo form, this problem is solved by the presence of "plug" and "valve" residues that simultaneously (yet reversibly) interrupt any wires reaching from the central glutamates to the cyto- or periplasm, resulting in several water molecules being trapped in the ligand binding site.

The patterns of ligand accumulation in equilibrium spontaneous-binding simulations suggest that the distribution of aromatic and charged side chains on the solvent-exposed sides of the protein, in particular the open side, is optimized for binding of charged aromatic ligands. In fact, the efflux of aromatic cations is particularly efficient due to a high barrier for spontaneous re-entry through the membrane. While aromatic cations and anions accumulated similarly at the solvent-exposed entry site, we note that most charged antibacterial agents are cations rather than anions. In addition, the negatively charged ligand-binding site will not accommodate an anion due to electrostatic repulsion: while a purely hydrophobic/aromatic driving force enables ligand entry up to 0.8 nm from the center (Figs 2B and 4C), the electrostatic attraction is required to access the binding site (Fig 2B).

The spontaneous entry simulations show how the ligand opens the collapsed channel of the apo state, and then rearranges the H-bonding patterns in the binding site to accommodate the bulky molecule. The same steric factor induces kinking of TM3 at the flexible GVG motif, a feature implicated in allosterically coordinating the conformational change and, potentially, elastic storage of the energy of binding. Ligand binding also stabilizes the binding channel, making it more accessible to water. Interestingly, the mechanism of ligand translocation by "tumbling" through the channel also hints at a role of aromatic residues in enhancing the kinetics of entry and exit.

Looking from a drug design perspective, the mechanism described here is challenging to disrupt: here, the sole presence of a ligand in the binding site promotes protonation of the central glutamates and the conformational change, suggesting that if a small molecule enters the binding site, it will also find its way out. One strategy assumed in the past was to design stapled peptides that compete with EmrE dimerization, thereby decreasing the concentration of functional dimers [28]. From a small-molecule point of view, though, the most robust strategy might rather be that using divalent ligands: even though EmrE is known to transport methyl viologen, a small divalent aromatic cation, through a proton antiport mechanism [49], one can

envision a molecule composed of an aromatic cation linked with a flexible chain to a group that is too bulky to pass through the channel. As a result, translocation of the catonic part would leave the bulky part behind, effectively blocking further transport. It remains to be seen, though, if such a system can be made functional against MDR bacteria.

## 4 Methods

### 4.1 System preparation

The EmrE protein structure refined by Ovchinnikov V. et al. [28, 37], based on the experimental structure (PDB ID: 3B5D) [50], was used as a initial structure. The construction of the full system has been carried out through CHARMM Membrane Builder [51]. Briefly, the protein with its ligand $PPh_4^+$ was embedded in a model lipid bacterial membrane composed of 70% (168) POPE and 30% (72)POPG molecules, solvated with 13167 water molecules and 103 $Na^+$ and 34 $Cl^-$ ions by using the Monte Carlo method. For the $PPh_4^+$ ligand, atomic charge calculations were performed using the Gaussian [52] program. The CHARMM36 forcefield [53] was used for bilayer and TIP3 model was used for water. Equilibration and later simulations were performed with the GROMACS 2018.4 and plumed 2.5.0 software [54, 55]. The system was equilibrated in seven steps with the procedure described in the CHARMM-GUI, where the energy of the system was first minimized, next equilibrated in 5 steps with constantly decreasing restraint forces applied to the system (see S1 Fig for a schematic description of simulation lengths and dependencies). The simulations were conducted with Nose-Hoover thermostat at 310 K and semi-isotropic Parrinello-Rahman barostat was used to kept pressure at 1 Bar. The velocity Verlet algorithm equations of motion were integrated with a time step of 2 fs. For a long-range electrostatic interactions Particle Mesh Ewald (PME) algorithm was used with a real-space cutoff of (10 Å) and Van der Waals and short-range Coulomb interactions were represented by using a smooth cutoff of (12 Å) with a switching radius of 10 Å. The system prepared in this way was then subjected to a 1 $\mu$s equilibrium simulation. Then, using the correlation-based cryo-EM fitting module recently implemented in Gromacs package, the protein was optimized for its density map [26] for extra 130 ns. The resulting refitted structure, as well as the parameters for $PPh_4^+$, are available at gitlab.com/KomBioMol/emre-setup.

### 4.2 Altered protonation states

To simulate specific protonation states of the two glutamic acids (main amino acids E14$^A$ and E14$^B$ responsible for the transport activity of the protein) in free energy simulations, additional hydrogens were added to each seeding frame obtained for the deprotonated variant, so that all four runs (umbrella sampling and metadynamics) were initialized from identical coordinates. Our in-house Gromologist library (gitlab.com/KomBioMol/gromologist) was used to automatize this process.

### 4.3 Simulations of spontaneous entry

The apo state of EmrE was prepared by removing the ligand from the system and then performing an equilibrium simulation of 1.75 $\mu$s. The obtained structure was closed to water on both sides of the protein, as shown in Fig 3A, 3C and 3D. Then, the Z-size of the box was increased by 1Å to randomly place 10 of each ligands in the resulting space. During the simulation, the applied pressure closed this gap, allowing ligands to diffuse freely. This process was repeated 3 times and for each ligand, and 1 $\mu$s long trajectories were obtained each time. Then, 3 structures closest to full entry into the binding site were extracted from the $PPh_4^+$ and $BPh_4^-$ systems and their parameters were swapped to produce another set of 1 $\mu$s long trajectories.

### 4.4 1D/2D metadynamics

To obtain the free energy profiles, we used two variants of well-tempered multiple walker metadynamics [56, 57] as implemented in the PLUMED package [55]. For the 1D variant, only the z-coordinate was used, defined as the z-component of the distance vector from the protein center of mass to the $PPh_4^+$ phosphorus atom. The ligand was restrained to a cylinder with a diameter of 1.5 nm and height of 6 nm from the center of the protein. In the 2D variant, the difference between the RMSD with respect to two symmetrically swapped domains (corresponding to the major conformational change) was used as an additional reaction coordinate. Overall, 20 walkers with 500 ns per walker (1D) or 26 walkers with 400 ns per walker (2D) were used for each of the four protonation states, yielding a total of 10 $\mu$s per protonation state per one free energy calculation. Initial structures were equally spaced along the Z-coordinate, and were taken either from a steered MD run (up to z = 1.1 nm) or from spontaneous entry simulations (z>1.1 nm). In the 2D variant, 13 walkers were initialized with the open side pointing upward and the other 13 downward; the latter were obtained by simple geometric manipulations, i.e. adequate rotation and relabeling of chains.

### 4.5 Umbrella sampling

25 frames with different ligand positions along the Z axis were created for umbrella sampling. They were obtained by using steered MD to pull the ligand through the membrane using the Gromacs package patched with Plumed. The ligand was restrained along the z-axis using a harmonic potential with a force constant of 500 kJ/nm$^2$, and simulated for 500 ns. After that, we unbiased all histograms separately for each ligand, using the weighted histogram analysis (WHAM) method [58] implemented in an in-house script (gitlab.com/KomBioMol/wham).

For ligand transition, the pathway was divided into 36 windows with a spacing of 0.05 nm (from -0.3 to 1.45 nm) and a force constant of 5000 kJ/nm$^2$, and 8 windows with a spacing of 0.2 nm (from 1.6 to 3.0 nm) and a force constant of 200 kJ/nm$^2$. For each window, 200 to 250 ns of trajectory was generated, totaling ca. 10 $\mu$s per protonation state per one free energy calculation, as in the case of metadynamics. The profiles were analyzed in the same manner as above.

Individual profiles from the above free energy methods were averaged to obtain the results shown in Fig 2, following an approach suggested by Laio and Gervasio [59]. These profiles, as well as their convergence, are shown in S4 Fig.

### 4.6 Grotthuss wire analysis

To identify continuous wires of h-bond-connected water molecules, we used an in-house script (gitlab.com/KomBioMol/proton_wire) that applies a graph search approach to find connections between "source" residues (here: the central glutamates) and a "target" (here: waters above or below the bilayer phosphate level for wires oriented up- and downward, respectively). Standard distance- and angle-based thresholds ($\leq$0.35 nm for acceptor-donor distance and $\geq$120deg for acceptor-proton-donor angle) were used for h-bond identification.

### 4.7 Alchemical calculation of pK$_a$

Calculations of pK$_a$ of glutamic acids were carried out using the Gromacs package patched with Plumed, with an in-house script for topology preparation (gitlab.com/KomBioMol/proton_alchemist). As a reference, a terminally capped glutamic acid was placed and restrained in the water phase with its alchemical states swapped, so that the system was neutral at all times,

and the resulting ΔG was identical to ΔΔG corresponding to the $pK_a$ shift. The $\Delta pK_a$s were calculated from the Henderson-Hasselbach equation using ΔG values obtained with Bennett Acceptance Ratio (BAR), and added to the reference value for free glutamic acid in solution (4.25).

$$\Delta G = -RTlnK_a$$

$$\Delta pKa = \frac{\Delta G}{2.303RT}$$

$$pKa_1 = pKa_2 + \frac{\Delta G}{2.303RT}$$

$$pKa_1 = pKa_2 + 0.174\Delta\Delta G$$

## 4.8 Weighted ensemble simulations

WESTPA (The Weighted Ensemble Simulation Toolkit with Parallelization and Analysis) is a high-performance Python framework for simulating rare events on an extended timescale with rigorous kinetics using the Huber-Kim weighted ensemble algorithm [60]. In order to sample the rare event of ligand entry into or escape from the protein, the reaction interval is divided into small bins that can be reached from neighboring ones almost immediately, during a single 10-ps iteration. Within each bin, a fixed number of simulations is carried out in each iteration, and individual runs are then merged or duplicated in a way that preserves the total probability. Initially, 33 such bins were defined, with higher numbers of simulations per bin in areas corresponding to predicted free energy barriers. These values were then adjusted depending on how far the ligand progressed and where it was trapped kinetically. Movies illustrating the respective pathways were generated with Molywood [61].

## Supporting information

**S1 Fig. Workflow of the refinements and simulations performed in this study.** See Methods for the details of all approaches used.
(TIF)

**S2 Fig. Alignment of the refined structure with the cryo-EM densities.** Using an early implementation of the cryo-EM module in the Gromacs package, we set the densfit-sigma parameter to 0.45 nm and densfit-k changed linearly from 0 to 10000 over 5 ns and remained at 10000 for the subsequent 95 ns. In the implementation, the "forward" model of the electron density is created by Gaussian-smoothing the atomic structure, and the forces are then applied to minimize the cross correlation between the simulated and actual density. This method allowed to obtain a structure with a proper spacing and orientation of helices, corresponding to the PDB entry 3B5D.
(TIF)

**S3 Fig. Alpha-carbon RMSD matrix between several experimental and computationally refined structures of EmrE, including the one shown in S2 Fig.** The lower part of the matrix contains RMSD values based on the full protein (except for the flexible C- and N-termini), while the upper part shows the corresponding values with the alpha carbons of the loops omitted. Legend: Ovchinnikov, the starting structure for this study taken from [28]; 7JK8, the most

recent (2021) NMR structure [29]; 3B5D, the CA-only low-resolution structure that was first reported in 2007 [26].
(TIF)

**S4 Fig. Individual free energy profiles used to construct the averaged profiles in Fig 2, as well as their convergence.** For the convergence analysis, data was divided into 6 batches and calculations were performed on cumulative datasets (batch 1, batch 1+2, . . .).
(TIF)

**S5 Fig. An alternative translocation pathway in which E14A protonation occurs before the conformational change.** This pathway is relatively more plausible under low cytoplasmic pH, and may represent an adaptation to variable pH conditions.
(TIF)

**S6 Fig. Free energy profiles for a single ligand passing through a pure POPE/POPG membrane.** The membrane was generated using the CHARMM-GUI webserver. The umbrella sampling protocol was applied to four different ligands (aromatic: cationic $PPh_4^+$, neutral $SiPh_4$, anionic $BPh_4^-$, and non-aromatic hydrophobic cation $NMe_4^+$) to create free energy profiles as a function of the z coordinate. The profiles indicate that charged ligands preferentially reside in the aqueous phase near the bilayer surface, the anionic ligand resides in the headgroup region and can likely pass through the membrane in a spontaneous manner, while the neutral tetraphenylsilane shows sizeable affinity for the bilayer interior.
(TIF)

**S7 Fig. The relative z-coordinates of ligands during the 1-$\mu$s charge-swap simulations.** 3-channel ligand structures (distance from to the binding site between 1.0 and 1.2 nm) were taken from the equilibrium simulation and six simulations of 1 $\mu$s each were performed for 3 systems with $PPh_4^+$, and 3 for $BPh_4^-$. The performed simulations were run as "as it is" or by changing the charges while maintaining the geometry. Of the 3 systems containing $PPh_4^+$ (yellow / orange / brown lines), all remained stably related to the channel entry, mainly going deeper (up to 0.8 nm) into the channel. At the same time, the least bound $BPh_4^-$ dissociated in one of the systems (purple line), while the other two progressed deeper into the binding channel, as did their cationic counterparts (blue / turquoise lines).
(TIF)

**S8 Fig. Visualization of the spontaneous ligand entry into the active site.** Simulations obtained from Westpa provide an insight into the key amino acids involved in the ligand entrance into the binding site, here visualized at three crucial stages (Z = 1.1 / 0.8 / 0.0) where the ligand dwells due to the presence of free energy barriers. The amino acids actively involved in the transport mechanism are shown: E14, Y60, Y40, F44 and W63.
(TIF)

**S9 Fig. Projection of the spontaneous ligand exit trajectory on the two main principal components describing correlated changes in H-bonding patterns.** The short trajectory from Westpa was merged with a follow-up 100 ns equilibration.
(TIF)

**S1 Table. The p$K_a$s obtained from the alchemical free energy calculations (described in Methods).** The table shows the p$K_a$ values obtained for a system with the ligand in the active site (Ligand-bound) and without it (Apo form). The vertical column labels (E14A/E14B) denote simulations in which a proton was alchemically exchanged between either E14A or

E14B embedded in the protein and a capped free GLU residue placed in the water phase, as described in the Methods.
(TIF)

**S1 Movie. The dominant ligand transport pathway, as postulated in Fig 2.** After entering the binding site, the ligand accelerates the conformational change that subsequently allows for preferential protonation of E14$^B$. A second protonation event allows for rapid escape of the ligand, leading to the apo state.
(MP4)

**S2 Movie. Water wires and desolvation in the apo state.** The inset shows Gaussian-smoothed probability of Grotthus wire formation with the top (orange) or bottom (blue) compartment as a function of time. After ca. 1 microsecond, the rearrangement of M21 and W63 creates a watertight seal that disrupts wires extending to the binding site, thereby preventing futile proton transfer.
(MP4)

**S3 Movie. Spontaneous ligand entry (left) and escape (right) trajectories obtained from Westpa, along with independently calculated free energy profiles for the fully deprotonated (orange) and doubly protonated (blue) systems.** Residues that come into contact with the P4P ligand are shown in yellow.
(MP4)

## Acknowledgments

This research was supported in part by PL-Grid Infrastructure and the TASK computational centre.

## Author Contributions

**Conceptualization:** Maciej Bagiński, Jacek Czub, Miłosz Wieczór.

**Investigation:** Jakub Jurasz, Miłosz Wieczór.

**Methodology:** Jakub Jurasz, Jacek Czub, Miłosz Wieczór.

**Project administration:** Maciej Bagiński, Jacek Czub.

**Supervision:** Maciej Bagiński, Miłosz Wieczór.

**Visualization:** Jakub Jurasz, Miłosz Wieczór.

**Writing – original draft:** Jakub Jurasz, Miłosz Wieczór.

**Writing – review & editing:** Maciej Bagiński, Jacek Czub.

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
