## [Decision Letter · Decision Letter 0]

9 Aug 2021

Dear Mr. Wieczor,

Thank you very much for submitting your manuscript "Molecular mechanism of proton-coupled ligand translocation by the bacterial efflux pump EmrE" for consideration at PLOS Computational Biology.

As with all papers reviewed by the journal, your manuscript was reviewed by members of the editorial board and by several independent reviewers. The reviewers overall appreciated the manuscript, but they had several substantial concerns.  We would therefore encourage you to revise the manuscript in a manner that addresses their concerns and submit it for reconsideration.

We cannot make any decision about publication until we have seen the revised manuscript and your response to the reviewers' comments. Your revised manuscript is also likely to be sent to reviewers for further evaluation.

Sincerely,

Peter M Kasson

Associate Editor

PLOS Computational Biology

Nir Ben-Tal

Deputy Editor

PLOS Computational Biology

Reviewer's Responses to Questions

**Comments to the Authors:**

Reviewer #1: The paper from Jurasz et al. details a comprehensive computational investigation of the proton-coupled ligand transport process by the EmrE dimer. The authors carry out long equilibrium simulations as well as umbrella sampling and metadynamics for free-energy calculations. From these, they propose two possible series of events for the transport cycle.

I generally enjoyed the many different computational approaches brought to bear on the question of transport by EmrE. I have a few questions and suggestions.

1a) Why were both umbrella sampling and metadynamics used for free-energy calculations? And how were they combined? I don't think that you can just average distinct free-energy profiles. The authors should provide a mathematically rigorous approach for combining (if they still want to do so) and/or provide individual profiles in the SI.

1b) Related, the authors should provide plots indicating the convergence of the free-energy profiles.

2) I find Figures 2C and 2D too busy and difficult to interpret. I don't think the faded profiles, for example, help. Can the authors find a way to simplify the presentation?

3) Can the authors propose new experiments that could test some of their conclusions?

Reviewer #2: Jurasz and coworkers present their computational studies for the multi-drug resistance transporter EmrE. The manuscript presents the results across a broad simulation campaign to work through the molecular mechanism by which substrate translocation occurs. I think that this work is a good fit for PLOS Comp. Bio. following revision to emphasize how the findings fit into the larger EmrE literature.

In the introduction, it is stated that “To date, the transport mechanism of EmrE has not been unambiguously established on the atomic level, nor has a high-quality structure been obtained experimentally, a common issue among small flexible proteins”. However, a new NMR based structure was released earlier this year with PDBID 7JK8 (reference 46). I fully realize that this project was likely started before this structure was made available, but the text needs to be changed to reflect this reality, and the appropriate comparisons need to be made to provide the reader with the full context for these results.

Methodological question: Where did you obtain the starting PDB structure? The Ovchinnikov model isn’t published alongside their paper, and so a reader can’t really assess for themselves how big the structural changes you imposed by refitting to CryoEM are. The other recent re-refined model (reference 27) used a similar refit based on CryoEM, and is available in the SI of the paper. It would be interesting to know how different the two models are, especially when compared to the recent NMR structure. RMSDs would be easy to calculate, and can help to assess how similar or different the models are from one another. It would also be valuable for the wider community if the EmrE model generated were to be released along with the SI.

Figure 2 is full of information! Possibly too much to be put into one figure. Particularly C and D are hard to understand, since it looks like there are 3 binding events under consideration, when it is clear that only 1 TPP binds. In panel D, E14A protonates twice! I think in this case, I might demote C and D to SI panels, and instead use simplified energy level diagrams like you might see in QM papers (eg. Fig. 5 from https://www.mdpi.com/2073-4344/9/11/887/htm, which happened to be the first one Google images came up with) to highlight the alternative energy landscapes in TPP translocation.

In Figure 3A, what were the parameters used to assess the presence or absence of a water wire? Typically, a geometric cutoff for both heavy-atom distance and angle linearity are used, but I can’t find where they are specified. I’ll take your word that Figure 3D shows what you say it does, but it is too small for a viewer to see the residues clearly.

Methods notes: Please specify the GROMACS version used for simulation (e.g. 5.1, 2020, etc). Sometimes there are bugs that are discovered later, and not having the version number available makes it unclear to readers in the future if these simulations might have this bug present or not. I assume the protein and TPP+ parameters also come from the CHARMM36 family. Citations should be included for maximal clarity, and consider including the parameters used for TPP as an SI component.

Do I understand correctly that the pKa calculations were carried out twice? Once moving the proton from an external glutamate to E14A, and once moving the proton from the external glutamate to E14B? Experimentally, the pKas for both glutamates are known (abstract of reference 38), and while E14A in the apo form and liganded forms bracket the experimental values, the manuscript would be strengthened by actually making the comparison within the text. Qualitatively, the results I see in the table S1 agree with prior modeling studies that measured delta pKa (reference 27), identifying E14B as the site that has the higher pKa, and therefore is protonated more frequently. I would, however, take another look at the labeling surrounding table S1, since the labels are currently insufficient to clarify what was actually calculated.

Combining pKa measurements here along with prior experiment, it seems clear that one of the glutamates is effectively constitutively protonated, since its pKa is below physiological pH. I think it would be useful to comment on this, given that the glutamates in the TPP bound simulations are both deprotonated.

Finally, the discussion around Figure 2 would be strengthened if it included discussion of alternative conformational exchange cycles for EmrE. It may be possible that the 2:1 stoichiometry is not a hard-and-fast rule (see 10.1073/pnas.1708671114, particularly its figure 5). If I read the order of events correctly, you have TPP binding first, followed by two protonation events. Clearly, the free energy change overall if all three of those had to happen each transport cycle would be the same. But what if, for instance, one proton remained bound during the full transport cycle? Would that be expected to be a faster process?

**Have the authors made all data and (if applicable) computational code underlying the findings in their manuscript fully available?**

Reviewer #1: None

Reviewer #2: **No: **I don't see a link specified in the manuscript to the underlying data. A tar.gz file uploaded to zenodo would satisfy the PLOS Data Policy requirement, I think.

PLOS authors have the option to publish the peer review history of their article (what does this mean?). If published, this will include your full peer review and any attached files.

Reviewer #1: No

Reviewer #2: No
---

## [Decision Letter · Decision Letter 1]

15 Sep 2021

Dear Mr. Wieczor,

We are pleased to inform you that your manuscript 'Molecular mechanism of proton-coupled ligand translocation by the bacterial efflux pump EmrE' has been provisionally accepted for publication in PLOS Computational Biology.

Best regards,

Peter M Kasson

Associate Editor

PLOS Computational Biology

Nir Ben-Tal

Deputy Editor

PLOS Computational Biology

Reviewer's Responses to Questions

**Comments to the Authors:**

Reviewer #1: Looks good to me! Upon reflection, I realized there's no difference between averaging the PMFs and the (mean) forces themselves.

Reviewer #2: The authors have addressed my primary concerns regarding the state of the manuscript, and have placed the data needed to recapitulate the key elements of the work online in an accessible form. I recommend publication.

**Have the authors made all data and (if applicable) computational code underlying the findings in their manuscript fully available?**

Reviewer #1: None

Reviewer #2: Yes

PLOS authors have the option to publish the peer review history of their article (what does this mean?). If published, this will include your full peer review and any attached files.

Reviewer #1: No

Reviewer #2: No

---

## [Editor Report · Acceptance letter]

30 Sep 2021

PCOMPBIOL-D-21-01276R1 

Molecular mechanism of proton-coupled ligand translocation by the bacterial efflux pump EmrE

Dear Dr Wieczor,

I am pleased to inform you that your manuscript has been formally accepted for publication in PLOS Computational Biology. Your manuscript is now with our production department and you will be notified of the publication date in due course.

With kind regards,

Olena Szabo
